# MicroRNAs Regulate Ca^2+^ Homeostasis in Murine Embryonic Stem Cells

**DOI:** 10.3390/cells12151957

**Published:** 2023-07-28

**Authors:** Kimberley M. Reid, Juan Miguel Sanchez-Nieto, Sandra Terrasse, Danilo Faccenda, Barbara Pernaute, Michelangelo Campanella, Tristan A. Rodriguez, Bradley S. Cobb

**Affiliations:** 1Department of Comparative Biomedical Sciences, Royal Veterinary College, University of London, 4 Royal College Street, London NW1 0TU, UK; kimberley.reid@ucl.ac.uk (K.M.R.);; 2National Heart and Lung Institute, Imperial College London, Hammersmith Hospital Campus, Du Cane Road, London W12 0NN, UKtristan.rodriguez@imperial.ac.uk (T.A.R.); 3Department of Clinical, Pharmaceutical and Biological Sciences, University of Hertfordshire, Hatfield AL10 9AB, UK; d.faccenda@herts.ac.uk; 4Centre for Clinical Pharmacology and Precision Medicine, William Harvey Research Institute, Queen Mary University of London, London E1 4NS, UK; m.campanella@qmul.ac.uk; 5University College London Consortium for Mitochondrial Research, London WC1E 6BT, UK; 6Institute Gustave Roussy, 94800 Villejuif, France; 7Department of Biomedical Sciences, University of Padua, 35122 Padua, Italy

**Keywords:** miRNAs, dicer, embryonic stem cells, Ca^2+^ regulation, stress response, IP_3_ receptors

## Abstract

MicroRNAs (miRNAs) are important regulators of embryonic stem cell (ESC) biology, and their study has identified key regulatory mechanisms. To find novel pathways regulated by miRNAs in ESCs, we undertook a bioinformatics analysis of gene pathways differently expressed in the absence of miRNAs due to the deletion of *Dicer*, which encodes an RNase that is essential for the synthesis of miRNAs. One pathway that stood out was Ca^2+^ signaling. Interestingly, we found that *Dicer*^−/−^ ESCs had no difference in basal cytoplasmic Ca^2+^ levels but were hyperresponsive when Ca^2+^ import into the endoplasmic reticulum (ER) was blocked by thapsigargin. Remarkably, the increased Ca^2+^ response to thapsigargin in ESCs resulted in almost no increase in apoptosis and no differences in stress response pathways, despite the importance of miRNAs in the stress response of other cell types. The increased Ca^2+^ response in *Dicer^−^*^/*−*^ ESCs was also observed during purinergic receptor activation, demonstrating a physiological role for the miRNA regulation of Ca^2+^ signaling pathways. In examining the mechanism of increased Ca^2+^ responsiveness to thapsigargin, neither store-operated Ca^2+^ entry nor Ca^2+^ clearance mechanisms from the cytoplasm appeared to be involved. Rather, it appeared to involve an increase in the expression of one isoform of the IP_3_ receptors (*Itpr2*). miRNA regulation of *Itpr2* expression primarily appeared to be indirect, with transcriptional regulation playing a major role. Therefore, the miRNA regulation of *Itpr2* expression offers a unique mechanism to regulate Ca^2+^ signaling pathways in the physiology of pluripotent stem cells.

## 1. Introduction

miRNAs play critical roles throughout mammalian development. These small RNAs of approximately 22 nucleotides regulate gene expression post-transcriptionally by targeting sequences in mRNAs and facilitating translation inhibition and message destabilization [1]. miRNAs are encoded within the genome as a part of larger transcripts, which are processed to maturity by protein complexes containing the essential RNases Drosha and Dicer [2]. The loss of miRNA synthesis in the early embryo through a genetic knockout of *Dicer* results in patterning defects in addition to lethality due to a massive apoptosis of embryonic cells at the post-implantation stage [3,4]. Therefore, an important question is how miRNAs function in regulating the physiology and developmental programs of the early embryo. Embryonic cell lines have provided an important means to study the mechanisms of miRNA function in early development. Their ability to differentiate into all cell types of the body has generated significant interest in understanding the mechanisms that control their pluripotency and overall physiology. Of these, embryonic stem cells (ESCs) are the most studied, and many functions have been found for miRNAs in regulating their biology.

Murine ESCs can survive in the absence of miRNAs but have a slower growth rate than control cells [5]. Interestingly, they are unable to exit the pluripotent state and differentiate [6], and much is known about the miRNA mechanisms regulating the expression of pluripotency transcription factors and ESC differentiation [7,8,9,10,11]. miRNAs also regulate other aspects of ESC biology. miRNAs of the miR-290/302 family (the most abundant miRNA family in ESCs) suppress cell cycle inhibitors and facilitate the shortened cell cycle that occurs in ESCs [12]. These miRNAs also regulate metabolism towards aerobic glycolysis, which is essential for ESC growth and pluripotency [13]. Additionally, miRNAs have been implicated in regulating the stress response [14] and the bivalent epigenetic regulation of developmental genes, which is specific to stem cells and in which chromatin within these genes contains both repressive and active modifications [15]. Therefore, miRNAs are key regulators of multiple functions in ESCs, and studying their roles has led to a greater understanding of the regulatory pathways involved in ESC physiology.

We wished to further examine miRNA function in ESCs and discover additional regulatory mechanisms in ESC physiology. Therefore, we undertook a bioinformatic analysis of genes differentially expressed in *Dicer^−^*^/*−*^ ESCs to identify novel pathways regulated by miRNAs. One group of genes that stood out in this analysis was those involved in regulating Ca^2+^ signaling. Regulating Ca^2+^ levels plays a central role in the physiology and survival of cells [16]. In addition, Ca^2+^ signaling through calcineurin and NFAT is important for the differentiation of ESCs [17], and it plays an essential role in the early development of vertebrate embryos [18]. Furthermore, there is a developing body of literature suggesting that miRNAs can regulate the expression of genes involved in Ca^2+^ transport [19]. Therefore, we undertook an analysis of miRNA regulation of Ca^2+^ homeostasis in ESCs.

## 2. Materials and Methods

### 2.1. Cell Lines and Maintenance

*Dicer^fl^*^/*fl*^ ESCs containing the Cre-ERT2 recombinase gene targeted to the Rosa26 locus [20] were maintained in GMEM (Invitrogen, Waltham, MA, USA) media supplemented with 10% *v*/*v* Fetal Calf Serum, 50 mM β-mercaptoethanol, 1% *v*/*v* NaPyr, 1% *v*/*v* NEAA, 1% *v*/*v* L-Glutamine, 1% *v*/*v* Pen/Strep, 1 mM PD0325901 (ERK inhibitor), 3 mM CHIR99021 (GSK inhibitor) (all Sigma, St. Louis, MO, USA), and 1500 U LIF on gelatin-coated dishes. HEK293T cells were maintained in DMEM (Sigma), 10% *v*/*v* FCS, and 1% *v*/*v* Pen/Strep.

### 2.2. Microarray Analysis

For the microarray analysis, RNA from 3 independent experiments was extracted as described in “RNA extraction and qPCR”. Labeling and hybridization to Affimetrix Mouse Gene 1.0ST microarrays were performed at UCL Genomics at the Institute of Child Health. Normalization and statistical analysis were performed using Robust Multichip Average and then imported into Partek. Data can be accessed with the accession number GSE93725. The Babelomics data analysis platform [21] was used to analyze the predicted miRNA targeting those significantly up-regulated by a 1.2-fold change in *Dicer^−^*^/*−*^ ESCs (*p* < 0.05). The bioinformatics platform DAVID [22] was used to obtain gene ontology terms from that gene list.

### 2.3. Fura-2 Live Cell Imaging

Cells grown in glass bottom dishes were washed in Hanks Buffered Saline Solution (HBSS) (137 mM NaCl, 5.4 mM KCl, 0.25 mM Na_2_HPO_4_, 5.6 mM Glucose, 0.44 mM KH_2_PO_4_, 1 mM MgSO_4_, 4.2 mM NaHCO_3_, and 1.3 mM CaCl_2_, pH 7.4) before incubating with 4 µM Fura-2, AM (Invitrogen, Waltham, MA, USA) in HBSS for 30 min in the dark, followed by 3 washes in HBSS. Image acquisition was carried out using a Nikon UV microscope and Andor1 software. Emission after excitation at 340 nm and 380 nm was recorded every second. Ratios were then calculated at every time point for at least 30 cells in the specified field.

Cytoplasmic Ca^2+^ levels were determined using the method of [23]. ECSs were loaded with Fura-2, AM in Ca^2+^-free HBSS with 1 mM EGTA. The basal fluorescence 340/380 nm emission ratio was measured followed by the addition of 3 mM Ca^2+^ and 1 mM ionomycin (final concentrations) to obtain the maximum fluorescence ratio (Rmax). Finally, 1 mM EGTA (final concentration) was added to obtain the minimum fluorescence ratio (Rmin). Ca^2+^ levels were calculated using the following equation: [Ca^2+^] nM = Ca^2+^ Kd × (R – Rmin)/(Rmax – R) × Sf2/Sb2, where Ca^2+^ Kd in the cytosol is 225 nM, Sf2 is the 380 emission value from the basal fluorescence measurement above, Sb2 is the 380 emission value from the maximum fluorescence measurement above, and R is the average of the 340/380 emission ratios from the cells under the designated condition in an individual experiment.

### 2.4. Annexin V Staining

Cells were washed in PBS then harvested by trypsinization and resuspended in an Annexin binding buffer containing APC conjugated Annexin V (Ebioscience, San Diego, CA, USA) plus propidium iodide. Cells were then analyzed by flow cytometry, and percentages of positive cells were calculated using Flowjo software.

### 2.5. Western Blotting

Western blots were performed following standard procedures using whole-cell extracts. The Dicer antibody was from Santa Cruz, SERCA1 was from Thermo Fisher (Waltham, MA, USA), Itpr1 was from AbCam (Cambridge, UK), Itpr2 was from Novagen (Pretoria, South Africa), Itpr3 was from Merk (Rahway, NJ, USA), and β-Actin was from Sigma (St. Louis, MO, USA). All other antibodies were from Cell Signaling (Danvers, MA, USA).

### 2.6. RNA Extraction and qPCR

For mRNA, total RNA was isolated using an Rneasy kit (Qiagen, Hilden, Germany). cDNA was synthesized using SuperScript V reverse transcriptase (Invitrogen), and qPCR was performed with SYBRgreen PCR master mix (BioRad, Hercules, CA, USA) using GAPDH for normalization. For miRNA, total RNA including miRNAs was isolated using a miRNeasy kit (Qiagen). cDNA was synthesized using a miRCURY LNA Universal RT microRNA cDNA synthesis kit (Exiqon, Hovedstaden, Denmark), and qPCR was performed with the miRCURY LNA Universal RT microRNA qPCR kit, using U5 RNA for normalization.

### 2.7. Plasmids and Transfection

Sequences encoding the *Atp2a1*, *Itpr1,2, and 3* 3′UTRs were amplified from genomic DNA and cloned into the pGL3 SV40 control luciferase reporter (Promega, Madison, WI, USA). Primer sequences are available upon request. Plasmids were transfected into HEK293T using standard Ca_2_PO_4_ methodology and into ESCs using Lipofectamine 2000 (Invitrogen). Luciferase assays were performed using the Dual Luciferase™ reporter assay system (Promega) per the manufacturer’s instructions. miRNA mimics (MirVana from Thermo Fisher) were transfected into HEK293T cells as above and into ESCs using Lipofectamine RNAi max (Invitrogen).

### 2.8. Statistical Analysis

Statistical analysis of data was performed using Graphpad Prism (Graphpad Software, version 9, San Diego, CA, USA). Differences in Ca^2+^ levels and protein expression were analyzed using a two-way ANOVA. Analysis of Annexin V time-course experiments, where multiple wells from two experimental groups were treated and analyzed at set times, was carried out using a non-repeated-measures two-way ANOVA, followed by the Fisher’s exact Least Significant difference posthoc test. When analyzing relative data such as qPCR and luciferase assay, where control cells had a value of 1, a Wilcoxon Signed Rank test was performed. All data were plotted as the mean value with error bars displaying SEM. For all cases, *p*-values of less than 0.05 were accepted as significant.

## 3. Results

### 3.1. The Deletion of Dicer in ESCs Results in the Upregulation of Genes Involved in Ca^2+^ Signaling

To examine the roles of miRNAs in ESCs, we utilized a system previously optimized by us to study miRNA function in stem cell lines in which the deletion of *Dicer* could be induced to prevent miRNA synthesis [24]. This system was advantageous in that it allowed for the examination of the immediate effects of the loss of miRNAs and avoided any complications from secondary changes that could occur during the isolation of stable cell lines lacking *Dicer*. The deletion of *Dicer* was accomplished using ESCs containing floxed alleles of *Dicer* and a Cre-ERT2 fusion gene knocked into the ubiquitously expressed *Rosa26* locus [20]. The constitutively expressed Cre-ERT could be activated in these cells to delete *Dicer* by the addition of tamoxifen. Optimal conditions for the deletion of Dicer mRNA and protein and the loss of miRNAs were established in ESCs by treatment with varying concentrations of tamoxifen. After three days of treatment, 0.5 mM tamoxifen produced an approximately 20-fold loss of Dicer mRNA and protein and a significant reduction in selected miRNAs that are highly expressed in ESCs and the pre-implantation embryo [24] (Figure 1A,B). Tamoxifen was then removed to avoid any possible toxicity complications and the cells were cultured for two or three additional days. At these times, several miRNAs of the miR-290/302 family, which is the most abundantly expressed miRNA family in ESCs, remained significantly depleted, as well as miR-19a and 19b, which are also abundantly expressed in ESCs and the early embryo [24,25] (Figure 1C). Therefore, any cells that had escaped *Dicer* deletion that could have a selective growth advantage [6] did not overgrow the population during the timeframe of our analysis.

To further validate the loss of miRNA function in this system, we examined the expression by qPCR of *Rbl2*, *Lats2*, and *Cdk1a*, whose miRNA regulation was previously shown to be important in the differentiation and cell cycle regulation of ESCs [7,8,12]. These genes were all found to be similarly upregulated upon *Dicer* deletion, as has been reported in the above studies (Figure 1D). To gain a further understanding of the immediate effects of miRNA depletion, we performed gene expression arrays and found 1065 genes that were significantly upregulated (*p* < 0.05) by 1.2-fold or more (Appendix A). We then analyzed the 3′UTRs of these genes using Babelomics software (version 5) to determine the miRNAs that had an enrichment in target sites within these sequences. Interestingly, this analysis revealed the most significant enrichment for miRNA target sites for the miR-290/302, miR-20, and miR-19 families (Figure 1E), which are the most abundantly expressed miRNAs in ESCs and mouse embryos and account for approximately 70% of miRNA expression during early mouse embryogenesis [24]. Therefore, our transient deletion appeared to be a valid system to study miRNA regulation, and it allowed us to study the roles of miRNAs in ESCs while circumventing any secondary effects that may have emerged through the repeated passaging required for the establishment and maintenance of stable *Dicer^−^*^/*−*^ lines.

### 3.2. Dicer^−/−^ ESCs Display Increased Cytoplasmic Ca^2+^ Levels upon Thapsigargin Treatment

We next pursued the regulatory pathways that are disrupted upon transient miRNA depletion using the DAVID functional annotation tool. A total of 64 terms were found to have an enrichment of 2-fold or higher, and these could be broadly broken down into 7 categories—28 terms related to organogenesis, 14 related to signaling, 9 related to adhesion and cell migration, 5 related to proliferation, 2 related to cell commitment and patterning, 2 related to Ca^2+^ regulation, and 4 outside these categories (Appendix A). Of these categories, Ca^2+^ signaling stood out as a novel pathway not previously known to be regulated by miRNAs in ESCs. Therefore, we decided to investigate the roles of miRNAs in regulating Ca^2+^ signaling in ESCs. To analyze Ca^2+^ signaling, cytoplasmic Ca^2+^ levels were examined using live-cell imaging with the ratiometric dye Fura-2, AM. The 340/380 nm-emission ratio was analyzed over a time course of 4 min, and no difference was found in the basal Ca^2+^ levels between *Dicer* deleted (*Dicer*^−/−^) and undeleted (*Dicer^fl^*^/*fl*^) ESCs (Figure 2). Since the endoplasmic reticulum (ER) is the major storage site for Ca^2+^ in ESCs [26], we asked if perturbations in Ca^2+^ transport across the ER membrane would differentially affect cytoplasmic Ca^2+^ levels in *Dicer^−^*^/*−*^ ESCs. Thapsigargin inhibits Ca^2+^ uptake into the ER by blocking the sarcoendoplasmic reticulum Ca^2+^—ATPase (SERCA), which results in the release of Ca^2+^ into the cytoplasm through ER Ca^2+^ export transporters. Therefore, we tested if the *Dicer*^−/−^ and *Dicer^fl^*^/*fl*^ ESCs were differentially sensitive to the addition of thapsigargin and found that *Dicer*^−/−^ ESCs had an increased cytoplasmic Ca^2+^ response (Figure 2). The difference in response time in these experiments was not reproducible and was a technical issue of how fast the addition of the HBSS solution containing thapsigargin could mix with buffer in the dish and diffuse into the cells.

### 3.3. miRNA Regulation of Ca^2+^ Homeostasis Plays Virtually No Role in the Stress Response of ESCs

Increases in cytoplasmic Ca^2+^ levels can lead to apoptosis by inducing mitochondria to release cytochrome C and stimulate a cascade of caspase activation, leading to the cleavage and activation of caspase 3 [27]. Therefore, we tested if the increased levels of cytoplasmic Ca^2+^ in *Dicer*^−/−^ ESCs treated with thapsigargin enhanced apoptosis. At 1 μM, thapsigargin killed both *Dicer^fl^*^/*fl*^ and *Dicer^−^*^/*−*^ ESCs after only a few hours of exposure. Therefore, we tested apoptosis levels by Annexin V staining using a reduced concentration of thapsigargin at 200 nM. This gave an overall reduced Ca^2+^ response with fewer cells responding. However, there was a greater number of responsive cells in *Dicer^−^*^/*−*^ ESCs (Appendix A). After 8 h of 200nM thapsigargin treatment, there was no difference in apoptosis between *Dicer^fl^*^/*fl*^ and *Dicer^−^*^/*−*^ ESCs, nor was there a difference at 48 h, when most of the cells in both cell types stained positive for Annexin V. However, after 24 h of treatment, there was a slight but statistically significant increase in apoptosis in *Dicer*^−/−^ ESCs (Figure 3A). Interestingly, this increased sensitivity did not correlate with an increase in activated caspase 3 (Figure 3B). If anything, it appeared to lead to a decrease, but repeat experiments revealed that this difference was not statistically significant. Therefore, the difference in thapsigargin sensitivity after 8 h of exposure appeared to result from alternative mechanisms unrelated to the Ca^2+^-induced release of mitochondrial cytochrome C and activation of the caspase cascade.

Thapsigargin also induces canonical stress response pathways by stimulating the unfolded protein stress response by reducing Ca^2+^ levels in the ER [28,29]. This decreases the activity of Ca^2+^ binding chaperones such as BiP that are involved in protein folding and leads to the phosphorylation of eIF2α by PERK, which reduces the translation of most cellular proteins and gives the cell time to recover. However, if the stress is too severe, proapoptotic proteins such as CHOP become expressed and lead to apoptosis. Since miRNAs have been implicated in regulating the general stress response mechanisms of cells [30], we examined their roles in the unfolded protein response of ESCs by analyzing the levels and phosphorylation of the above key proteins. Interestingly, when compared to control cells, the expression of CHOP and BiP as well as the phosphorylation of eIF2α appeared to be attenuated upon miRNA depletion. However, densitometric analysis of this and repeat experiments revealed that the differences were not significant (Figure 3B). Therefore, miRNAs did not appear to regulate the unfolded protein response in ESCs and, in fact, they appeared to play insignificant roles in the overall stress response to thapsigargin of ESCs.

### 3.4. miRNAs Regulate Cytoplasmic Ca^2+^ Levels upon the Stimulation of Purinergic Receptors

A key question is whether miRNAs regulate Ca^2+^ levels under normal physiological conditions that do not involve pharmacological inhibitors of Ca^2+^ transport. Therefore, we examined if miRNAs could regulate normal signaling mechanisms that lead to increases in cytoplasmic Ca^2+^ levels. Exogenous ATP stimulates Ca^2+^ release in ESCs by binding to purinergic receptors and, just as with thapsigargin treatment, *Dicer^−^*^/*−*^ ESCs had an increased cytoplasmic Ca^2+^ response to ATP stimulation (Figure 4). Therefore, miRNAs are important in regulating the Ca^2+^ response of physiological signaling events of ESCs, which could play a role in ESC pluripotency and differentiation mechanisms.

### 3.5. The miRNA Regulation of Ca^2+^ Homeostasis Correlates with an Increased Expression of the IP_3_ Receptor 2 (Itpr2)

To test if the enhanced sensitivity of *Dicer^−^*^/*−*^ ESCs to thapsigargin was due to effects on Ca^2+^ transport across the ER membrane or from extracellular sources through store-operated Ca^2+^ entry, we treated ESCs with thapsigargin in the absence of extracellular Ca^2+^ using HBSS without Ca^2+^. Once again, *Dicer^−^*^/*−*^ ESCs displayed an increased cytoplasmic Ca^2+^ response (Figure 5A), indicating that mechanisms other than store-operated Ca^2+^ entry were important. Furthermore, the reduction of cytoplasmic Ca^2+^ levels over time in these experiments was nearly identical between *Dicer^fl^*^/*fl*^ and *Dicer^−^*^/*−*^ ESCs. Therefore, the increased Ca^2+^ response of *Dicer^−^*^/*−*^ ESCs appeared not to result from differences in cytoplasmic Ca^2+^ clearance mechanisms involving the export of Ca^2+^ through plasma membrane transporters or its sequestration into other organelles. Instead, the differences appeared to be primarily from effects on Ca^2+^ release from the ER.

One possible explanation for the miRNA-mediated buffering of Ca^2+^ levels in the cytoplasm could be through their regulation of Ca^2+^ transporters in the ER. Ca^2+^ release from the ER is mediated by the stimulation of ryanodine and inositol trisphosphate (IP_3_) receptors in the ER membrane, whereas import into the ER is mediated by SERCA, as stated above. ESCs have very low expression of ryanodine receptors [26] and, consistent with this, the stimulation of ryanodine receptors by the agonist caffeine gave a minimal Ca^2+^ response, which was very similar in *Dicer^fl^*^/*fl*^ and *Dicer^−^*^/*−*^ ESCs (Figure 5B). In contrast, IP_3_ receptors are expressed in ESCs, and one isoform, Itpr2 was found to be more highly expressed in *Dicer^−^*^/*−*^ ESCs. Likewise, one isoform of SERCA (SERCA1 encoded by *Atp2a1*) had increased mRNA levels. However, the increase at the protein level was harder to observe due to the weak detection by the SERCA1 antibody (Figure 5C). Therefore, these data suggested that *Dicer^−^*^/*−*^ ESCs have an enhanced export capacity that can be compensated by uptake through SERCA under normal conditions. This would explain why the basal Ca^2+^ levels are stable in *Dicer^−^*^/*−*^ ESCs but enhanced when Ca^2+^ uptake into the ER is blocked by thapsigargin.

### 3.6. miRNAs Primarily Regulate the Level of Itpr2 Expression Indirectly in ESCs

To determine if miRNA regulation of *Itpr2* and *Atp2a1* involves a direct targeting of miRNAs to sequences in their 3′UTRs, luciferase reporters were constructed containing their 3′ UTRs and tested for miRNA regulation in ESCs. In addition, 3′ UTR reporters were also constructed for *Itpr1* and *Itpr3*. These were transfected into *Dicer^fl^*^/*fl*^ and *Dicer^−^*^/*−*^ ESCs along with a Renilla luciferase gene to normalize transfection efficiency. Comparing the ratio of the relative luciferase activities between *Dicer^fl^*^/*fl*^ and *Dicer^−^*^/*−*^ ESCs revealed that the 3′ UTRs of *Itpr2* and *Itpr1* imparted miRNA regulation on luciferase expression, whereas the 3′ UTRs of *Itpr3* and *Atp2a1* did not (Figure 6A). The lack of effect of the *Atp2a1* 3′ UTR contrasted with the effect of miRNAs on the endogenous gene. Therefore, other mechanisms must indirectly mediate miRNA regulation of the endogenous gene. Also in discordance was the miRNA regulation imparted by the *Itpr1* 3′ UTR as the endogenous gene appeared not to be regulated by miRNAs. Therefore, the effect of its 3′ UTR appeared to be context-dependent. Nevertheless, its regulation along with that of *Itpr2* was pursued to identify individual regulatory miRNAs. 

To identify relevant miRNAs, we hypothesized that important miRNAs would be abundantly expressed in ESCs. miRNAs can be grouped into seed families, which contain the same seed sequence of approximately seven nucleotides near the 5′ end of the miRNA. Canonical miRNA targeting typically requires near-perfect homology across the seed sequence but tolerates multiple mismatches throughout the remainder of the sequence; thus, miRNA seed families generally target the same messages. Therefore, we looked for predicted miRNA target sites of the most abundantly expressed miRNA seed families in ESCs using Targetscan (www.targetscan.org, accessed on 1 June 2022) [31]. The top 10 expressed miRNA seed families in ESCs, as determined by the numbers of sequence reads in the whole-genome sequencing of small RNAs [25], are listed in Figure 6B. These account for over 80% of the sequence reads of all miRNAs expressed in ESCs; thus, these were investigated as likely candidates for regulating ER Ca^2+^ transporter genes. Consistent with a lack of miRNA regulation of the *Itpr3* and *Atp2a1* reporters in ESCs, the 3′ UTRs of these genes did not contain any predicted target sites for these miRNA seed families. In contrast, the 3′ UTRs of *Itpr1* and *Itpr2* did contain predicted target sites for several of these miRNA seed families, several of which were highly conserved in vertebrate species in addition to others that were poorly conserved. Representative miRNAs of these seed families were tested for their ability to regulate the *Itpr1* and *Itpr2* 3′UTR reporters in co-transfection experiments using HEK293T cells. All the tested miRNAs were able to suppress the expression of reporters containing synthetic optimal target sites in their 3′ UTRs (Appendix A); yet, the only miRNA that could suppress the expression of either of the *Itpr* reporters was miR-92 on *Itpr1* (Figure 6C). The suppression of the *Itpr1* reporter by miR-92 primarily required its predicted target site as the mutation of this sequence in the reporter significantly relieved suppression by miR-92 (Appendix A). Interestingly, the miR-92 mimic had no effect on the expression of the endogenous *Itpr1* in ESCs (Appendix A), which was consistent with the lack of miRNA regulation of the endogenous *Itpr1*. Likewise, it was not able to attenuate the Ca^2+^ response in *Dicer^−^*^/*−*^ ESCs (Appendix A). Therefore, the differences in the miRNA regulation of the *Itpr1* 3′UTR reporter and endogenous gene could be due to differences in the overall tertiary structure of their RNAs and/or their association with RNA-binding proteins, which affected the accessibility of the target site.

Other miRNAs either had no effect on either reporter or, in some cases, enhanced reporter expression, which presumably occurred through indirect mechanisms. With *Itpr2*, none of these individual miRNAs or their combination could suppress its expression. Therefore, the specific miRNA(s) regulating this reporter in ESCs could not be determined. This left the possibility that either less abundant miRNAs could be important or some of the non-tested abundant miRNAs could mediate suppression through non-conventional target recognition that is not dependent on the seed sequence. Alternatively, miRNAs could function indirectly by regulating the expression of RNA-binding proteins that interact with the *Itpr2* 3′ UTR and impact message stability and translation.

In addition to the above regulatory mechanisms through the 3′ UTR of *Itpr2*, miRNAs might also regulate *Itpr2* expression indirectly through transcriptional means. Consistent with this was the observation that the miRNA effect on the expression of the endogenous *Itpr2* was much greater than their effect on the *Itpr2* 3′ UTR reporter. To analyze transcriptional activity, primary transcript levels were measured because primary transcripts are typically spliced rapidly, leading to their levels generally being determined by transcriptional activity. qPCR primers across the first intron of *Itpr2* as well as *Itpr1* and *Atp2a1* were used to analyze their primary transcript levels in *Dicer^fl^*^/*fl*^ and *Dicer^−^*^/*−*^ ESCs. In addition, primers spanning across adjacent exons were used to measure the mature mRNA levels (Figure 6D). Of these three genes, only *Itpr2* displayed an increase in the level of its primary transcript in *Dicer^−^*^/*−*^ ESCs. Therefore, miRNAs appeared to regulate *Itpr2* transcription. In contrast, there was no increase in the primary transcript levels of *Atp2a1* or *Itpr1*, which, with *Itpr1*, was consistent with the lack of an increase in the mature transcript. However, the lack of a miRNA effect on the primary transcript levels of *Atp2a1* indicated that the increase in the mature mRNA level was not due to transcriptional regulation. Therefore, this and its lack of miRNA regulation through its 3′ UTR indicated that an indirect mechanism of regulating message stability appeared to be important.

How can miRNAs regulate the transcription of *Itpr2* in ESCs? The most straightforward mechanism would be through the miRNA suppression of important transcriptional activators. However, little is known about the transcriptional regulation of *Itpr2*, and no transcriptional activators have been identified. In contrast, two transcriptional repressors have been shown to suppress *ITPR2* expression in human cells. BTB and CNC Homology 1 (BACH1) and retinoid X receptor alpha (RXRA) suppress the expression of *ITPR2* in HEK293T cells and primary human fibroblasts, respectively [32,33]. Even if the miRNA-regulated expression of these were important for the increased *Itpr2* expression in *Dicer^−^*^/*−*^ ESCs, they would have to be indirectly upregulated by miRNAs. However, neither of these two repressors was differentially expressed in *Dicer^−^*^/*−*^ ESCs (Figure 6E). Therefore, miRNAs must regulate other transcriptional regulators, and identifying these miRNA targets will require a better understanding of *Itpr2* transcriptional regulation.

## 4. Discussion

In this report, we have found that the miRNA regulation of gene expression plays a role in buffering cytoplasmic Ca^2+^ levels upon exposure to thapsigargin or through normal signaling mechanisms that stimulate the release of Ca^2+^ from the ER. miRNA regulation of the IP_3_ receptor gene *Itpr2* offers one possible mechanism for mediating Ca^2+^ buffering; thus, it will be important to learn how miRNAs regulate *Itpr2* expression in ESCs. In other cell types, miRNAs have been found to directly target sites in the *Itpr2* 3′ UTR and regulate *Itpr2* expression. miR-34a and -133a regulate *Itpr2* expression in T-cells and myocardium, respectively [34,35]. However, these miRNAs are barely expressed in ESCs [25]; thus, they are unlikely to be important regulators of *Itpr2* expression in ESCs. In contrast, miRNAs play an important role in the transcriptional regulation of *Itpr2* expression in ESCs. Therefore, it will be important to understand the miRNA regulation of key transcriptional activators. However, the transcriptional regulation of *Itpr2* has been little studied; thus, the miRNA regulation of such activators awaits a better understanding of *Itpr2* transcriptional regulation.

An even more interesting question from these studies is how the miRNA buffering of cytoplasmic Ca^2+^ levels impacts the biology of ESCs. Ca^2+^ is an essential and ubiquitous signaling ion that plays a vast array of roles in cellular physiology, and its deregulation is directly linked to disease [36]. High mitochondrial Ca^2+^ levels are known to cause mitochondrial swelling and dysfunction and can trigger the opening of the mitochondrial permeability transition pore that activates apoptotic cascades [27]. However, there was no difference in the thapsigargin-induced activation of the terminal caspase (caspase 3) between *Dicer^fl^*^/*fl*^ and *Dicer^−^*^/*−*^ ESCs. Therefore, the increased Ca^2+^ response of *Dicer^−^*^/*−*^ ESCs did not further activate this pathway and, thus, the slight increase in thapsigargin-induced apoptosis in *Dicer^−^*^/*−*^ ESCs that was observed only after 24 h of exposure must have been due to other mechanisms.

Survival can also be regulated by the activation of canonical stress response pathways, and miRNAs have been found to regulate multiple steps in these pathways [30]. However, we found no enhancement of canonical stress response pathways in the *Dicer*^−/−^ ESCs under the conditions tested. Therefore, either (1) ESCs are different from other cell types and do not utilize miRNAs to regulate the stress response pathways or (2) the regulation of these stress response pathways was overwhelmed by the stress conditions utilized in this study.

Ca^2+^ signaling has also been suggested to play roles during the early stages of vertebrate differentiation into the neural and cardiac lineages [37,38,39]. The increased Ca^2+^ response to ATP stimulation could indicate that other pathways that induce Ca^2+^ could also be influenced by miRNAs; thus, it is tempting to speculate that alterations in the Ca^2+^ response to signaling pathways in *Dicer*^−/−^ ESCs could possibly impact differentiation. Unfortunately, we were not able to test this possibility here for two reasons. First, *Dicer*^−/−^ cell lines fail to differentiate as miRNAs are required for the exit of pluripotency via the regulation of genes such as *Rbl2*, which is required for the methylation and stable repression of Oct4 expression and which we also found upregulated in our transient deletions [7,8]. Second, when we allowed *Dicer^fl^*^/*fl*^ cells to exit pluripotency and then added tamoxifen to delete *Dicer,* we observed massive cell death (data not shown) due to the importance of miRNAs in regulating cell survival during differentiation [4,24]. Therefore, the effects of miRNAs involved in these processes may mask the effects of any miRNAs in regulating Ca^2+^ signaling. Because of this, it will be important to identify individual miRNAs that mediate the regulation of Ca^2+^ levels and analyze the impact of their manipulation on Ca^2+^ signaling during the differentiation of ESCs.

## 5. Conclusions

This study establishes an important new role for miRNAs in buffering Ca^2+^ levels in the pluripotent state and identifies another potentially important regulatory mechanism for their biology. However, until individual miRNAs are identified, we cannot eliminate the possibility that other functions of Dicer could also be important in regulating the expression of genes involved in Ca^2+^ homeostasis in ESCs. Dicer can process other RNAs in the cell and affect the function of other proteins to which it forms a complex [40]. However, the mechanisms by which these other functions of Dicer could regulate gene expression are much less understood and, in other developmental systems that are affected by the deletion of *Dicer*, specific miRNAs have been found to be important for the phenotype. Therefore, miRNA synthesis appears to be the major role of Dicer in development, but these alternative mechanisms of Dicer may prove to be additionally important.

## Figures and Tables

**Figure 1 cells-12-01957-f001:**
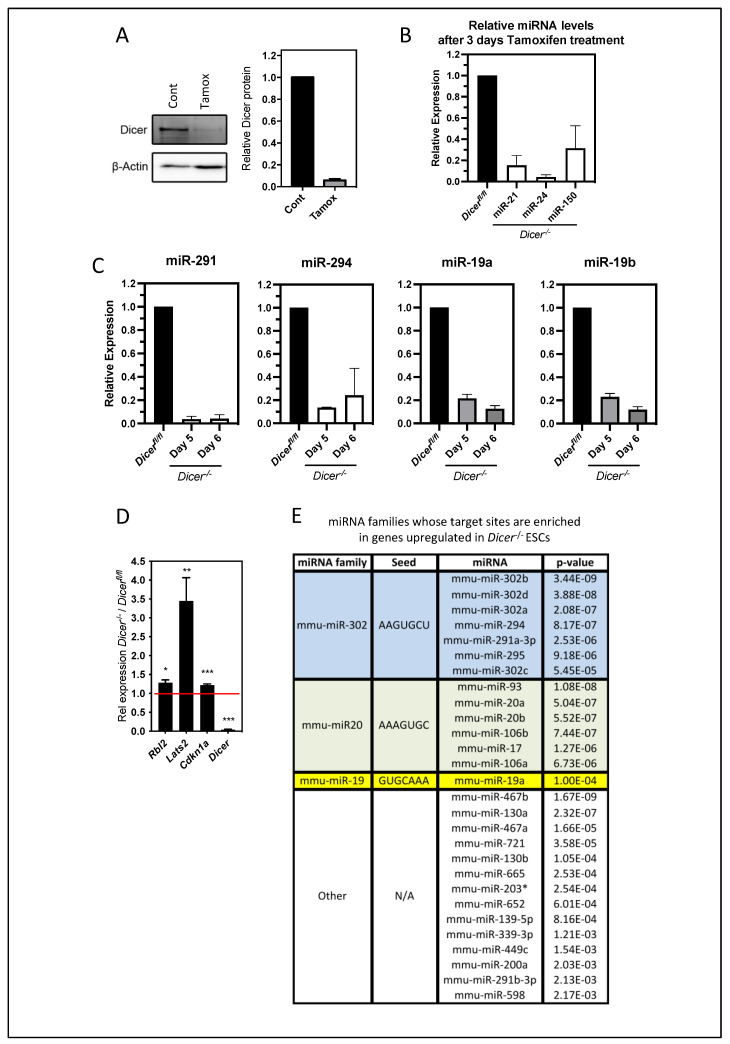
**Characterizing the deletion of *Dicer* and loss of miRNAs in ESCs.** (**A**) *Dicer^fl^*^/*fl*^ ESCs were treated with 0.5 mM tamoxifen for 3 days, and the loss of Dicer protein relative to β-actin was measured in Western blots of total cellular protein extracts. On the left is a representative experiment and on the right is the quantitation by densitometry from 10 independent experiments where Dicer proteins were determined relative to untreated cells. (**B**) qPCR analysis from total RNA revealed the loss of indicated miRNAs. Data are from three independent experiments repeated in duplicate. (**C**) Cells were cultured for an additional 2 or 3 days without tamoxifen, and the levels of the indicated members of the miR-290/302 and miR-19 families were determined by qPCR. Data are from two or three independent experiments repeated in duplicate. (**D**) Genes whose miRNA regulation was previously shown to be important for ESC differentiation and cell cycle regulation were similarly upregulated when compared to *Dicer*^−/−^ stable cell lines. qPCR data are from four independent experiments (*Rbl2* mean = 1.3-fold, *p* = 0.02; *Lats2* mean = 3.4-fold, *p* = 0.006; *Cdkn1a* mean = 1.2-fold, *p* = 0.0008; *Dicer* mean = 0.04-fold, *p* = 0.0003). (**E**) Table displaying the miRNA families and specific miRNAs belonging to these families that have a significant number of predicted targets enriched among genes upregulated in *Dicer^−^*^/*−*^ ESCs. Using the Babilomics platform, these studies identified the miRNAs of the miR-19, miR-20, and miR-290/302 families, (some of the most abundantly expressed miRNAs in ESCs and the early embryo [24,25]) as having a significantly enriched number of target sites within the upregulated genes. A two-tailed Fisher exact test was used to determine significance (adjusted *p* < 0.05), * *p* < 0.05, ** *p* < 0.01, *** *p* < 0.001.

**Figure 2 cells-12-01957-f002:**
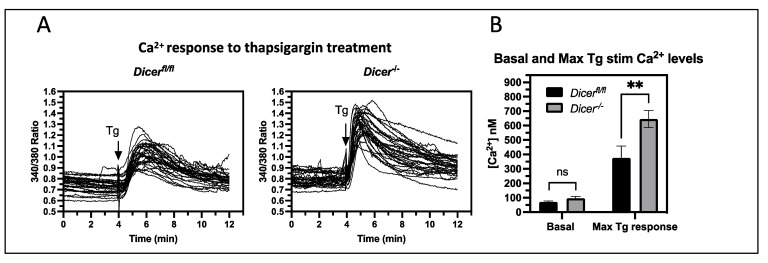
***Dicer^−^*^/*−*^ ESCs show increased cytoplasmic Ca^2+^ levels upon thapsigargin treatment**. *Dicer^fl^*^/*fl*^ and *Dicer*^−/−^ ESCs were loaded with 4 mM Fura-2, AM, and cytoplasmic Ca^2+^ levels were analyzed by fluorescence microscopy every second at 340 and 380 nm. The ratio of the emission at 510 nm from the two excitation wavelengths was plotted for individual cells (minimum of 30 cells per experiment). After 4 min, cells were treated with 1 μM thapsigargin (Tg). (**A**) Representative graphs from single experiments are shown. (**B**) The mean basal and peak thapsigargin stimulated ratios from individual experiments were used to calculate cytoplasmic Ca^2+^ levels (as described in the Materials and Methods section), and the bar chart shows the mean and SEM levels from six independent experiments. No difference was observed in the basal Ca^2+^ levels (ns—not significant). However, *Dicer^−^*^/*−*^ cells had an increased cytoplasmic Ca^2+^ response to thapsigargin (*p* = 0.0027), ** *p* < 0.01.

**Figure 3 cells-12-01957-f003:**
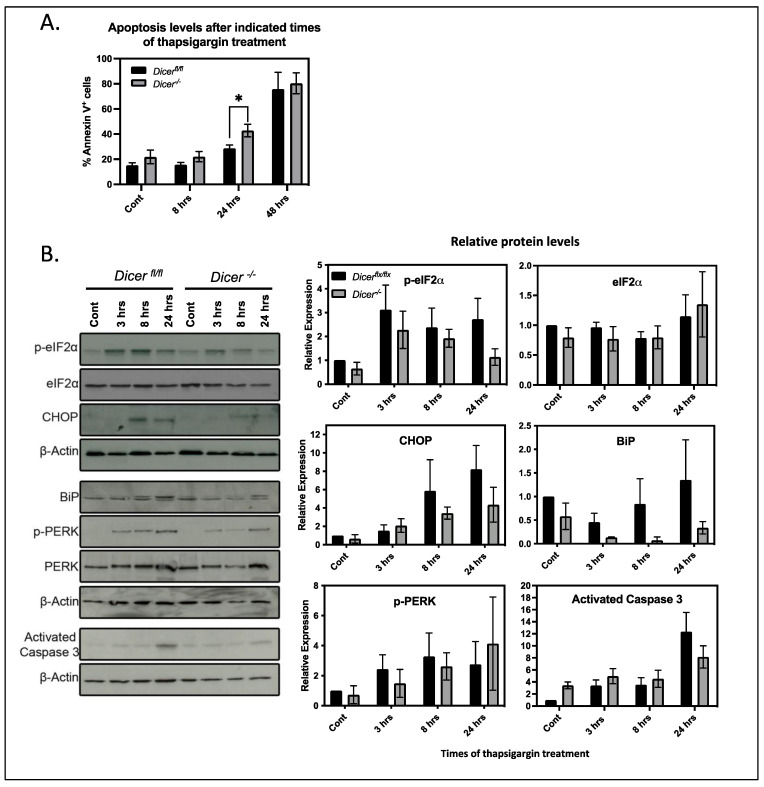
***Dicer*^−/−^ ESCs have virtually no difference in stress response pathways.** (**A**) *Dicer^fl^*^/*fl*^ and *Dicer*^−/−^ ESCs were treated for the indicated times with 200 nM thapsigargin. Apoptotic cells were determined by Annexin V staining directly after the indicated time of treatment. *Dicer^−^*^/*−*^ ESCs had a slight but significant increase in apoptotic cells after 24 h of thapsigargin treatment (*p* = 0.034) but showed no difference at 8 or 48 h of treatment. Data are from three independent experiments. (**B**) A total of 200 nM thapsigargin treatment for the indicated times did not induce an increase in the activation of the canonical stress pathways in *Dicer*^−/−^ ESCs as measured by the expression or phosphorylation levels of indicated proteins in Western blots. Representative Western blots of three independent experiments are presented, and the densitometric quantitation from repeat experiments normalized to b actin is shown on the right, * *p* < 0.05.

**Figure 4 cells-12-01957-f004:**
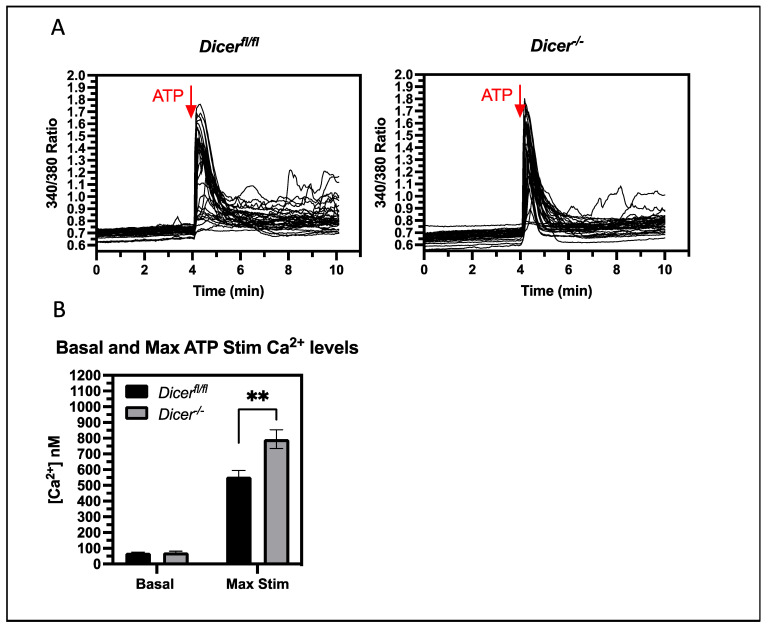
***Dicer^−^*^/*−*^ ESCs have an increased cytoplasmic Ca^2+^ response to ATP signaling**. *Dicer^fl^*^/*fl*^ and *Dicer*^−/−^ ESCs were loaded with Fura-2, AM and cytoplasmic Ca^2+^ levels were analyzed by fluorescence microscopy, as described in Figure 2. Basal levels were established over 4 min; then, cells were stimulated with 5 mM ATP. (**A**) Graphs from representative experiments are shown. (**B**) The bar chart shows the mean cytoplasmic Ca^2+^ levels and SEM values from four independent experiments. *Dicer^−^*^/*−*^ ESCs had an increased cytoplasmic Ca^2+^ response to ATP stimulation (*p* = 0.0011), ** *p* < 0.01.

**Figure 5 cells-12-01957-f005:**
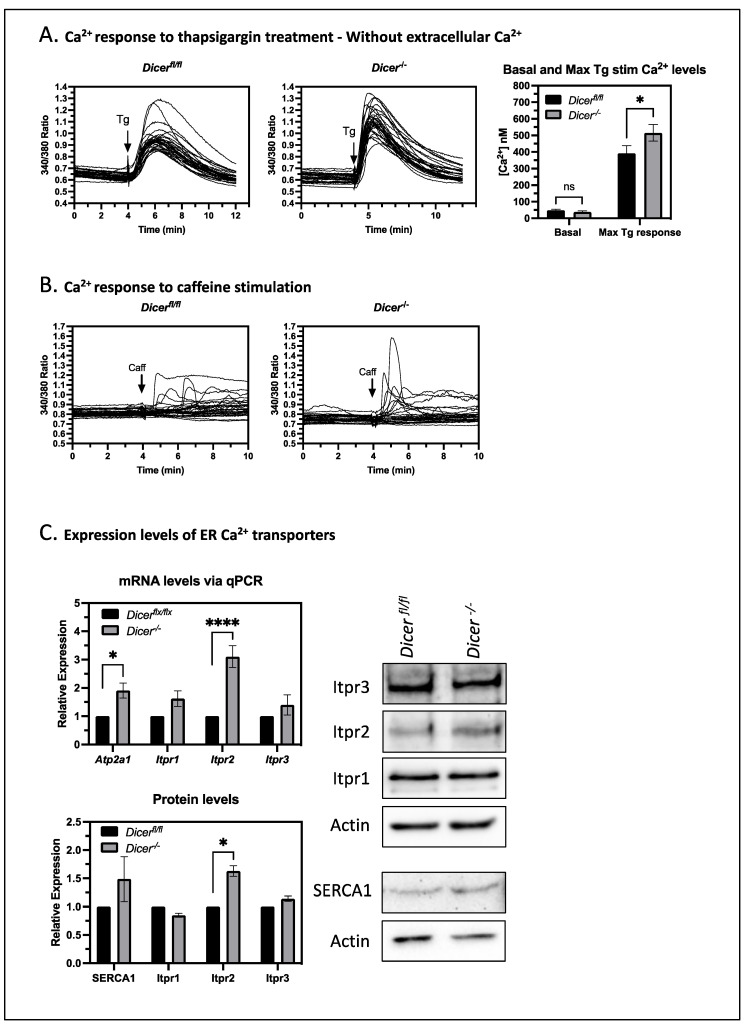
**The increased Ca^2+^ response in *Dicer^−^*^/*−*^ ESCs correlates with an increased expression of the IP_3_ receptor *Itpr2*.** (**A**) To examine the effect of store-operated Ca^2+^ entry on cytoplasmic Ca^2+^ levels, Fura-2, AM fluorescent experiments were performed, as described in Figure 2 but without extracellular Ca^2+^. On the left are graphs from representative experiments and on the right is a bar chart showing the mean cytoplasmic Ca^2+^ levels and SEM values from six independent experiments. *Dicer^−^*^/*−*^ ESCs still had an increased cytoplasmic Ca^2+^ response to thapsigargin (*p* = 0.035), indicating that mechanisms other than store-operated Ca^2+^ entry were important. In addition, the similar recovery rates to basal levels indicated that the clearance mechanisms were similar. Therefore, it appeared that an increased release from the ER was important for the enhanced thapsigargin response of *Dicer^−^*^/*−*^ cells. (**B**) ER membrane ryanodine receptors appeared not to be important for the enhanced cytoplasmic Ca^2+^ response of *Dicer^−^*^/*−*^ ESCs. Their stimulation by caffeine elicited a limited response in only a few cells, which was similar in both *Dicer^fl^*^/*fl*^ and *Dicer^−^*^/*−*^ ESCs. Representative graphs from three independent experiments are shown. (**C**) *Dicer*^−/−^ ESCs expressed higher levels of the ER Ca^2+^ transporter genes *Itpr2* (*p* < 0.0001) and *Atp2a1* (encoding SERCA1) (*p* = 0.0212), as measured by qPCR. Likewise, the protein level of Itpr2 as measured in Western blots was increased (*p* = 0.0116). SERCA1 levels also appeared to be increased. However, the weak detection by the antibody gave variable results such that the difference was not significant (*p* = 0.0632). qPCR and Western data are each from three independent experiments, * *p* < 0.05, **** *p* < 0.0001.

**Figure 6 cells-12-01957-f006:**
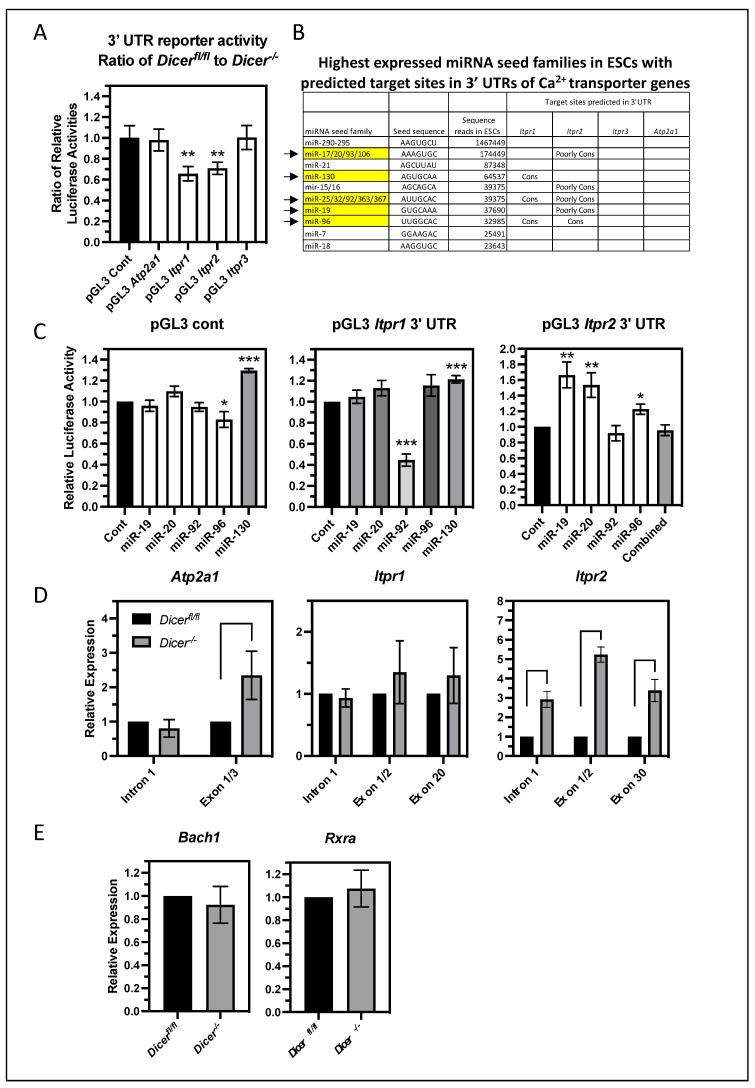
**miRNA regulation of the expression of *Itpr2* is complex and appears to primarily involve transcriptional mechanisms.** (**A**) The 3′ UTRs of *Itpr1* and *Itpr2* can elicit miRNA regulation in ESCs. Luciferase reporter genes containing the 3′ UTRs of the ER Ca^2+^ transporter genes were constructed and transfected into *Dicer^fl^*^/*fl*^ and *Dicer^−^*^/*−*^ ESCs along with a Renila luciferase reporter gene to control for transfection efficiency. The ratio of the relative luciferase activities as normalized to the parental control luciferase reporter is shown in the bar graph. Only the 3′ UTRs of *Itpr1* (*p* = 0.0026) and *Itpr2* (*p* = 0.0015) imparted miRNA regulation in ESCs. Data are from eight independent experiments. (**B**) The table shows the most abundant miRNA seed families in ESCs, as derived from the number of sequences reads in the RNA seq data of [25]. The presence of strongly and poorly conserved predicted target sites for these miRNAs as deduced from TargetScan is also shown. Highlighted in yellow and with arrows are the seed families for which representative members were tested for an effect in reporter assays with the *Itpr1* and *Itpr2* 3′UTR reporter genes. (**C**) The effect of representative miRNAs of the indicated miRNA seed families on these reporters was analyzed in HEK293T cells using miRNA mimics. The relative luciferase activity normalized to a control non-specific mimic is shown in the bar graphs. The only case of miRNA suppression was for miR-92 of the *Itpr1* 3′UTR reporter (*p* = 0.0010). All other miRNAs either had no effect or, in some cases, enhanced the expression of the reporters presumably through indirect effects. Data are from 12 independent experiments each with the exception of miR-96 on pGL3 cont *n* = 11 and pGL3 *Itpr2* 3′ UTR *n* = 7. (**D**) The miRNA regulation of *Itpr2* appeared to be mediated by transcriptional regulation. The relative increase in the primary transcript levels (as measured by qPCR primers to the first intron) (*p* < 0.0001) was reflected in the increase in the mature transcript levels, as measured by qPCR using primers to the indicated exon sequences (*p* < 0.0001). This was not the case for *Atp2a1* as there was no increase in its primary transcript. Data are from three independent experiments. (**E**) Expression of two known transcriptional repressors of *Itpr2, Bach1*, and *Rxra* is not regulated by miRNAs in ESCs. Mature mRNA levels were measured by qPCR using primers spanning adjacent exons. Data are from three independent experiments, * *p* < 0.05, ** *p* < 0.01, *** *p* < 0.001.

## Data Availability

Array data have been deposited in the GEO repository with the accession number GSE93725.

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
