# Peer review of "MicroRNAs Regulate Ca^2+^ Homeostasis in Murine Embryonic Stem Cells"

_cells, 2023, doi:10.3390/cells12151957_

Round 1
Reviewer 1 Report
This is a review of manuscript cells-2515614 “microRNAs regulate Ca2+ homeostasis in murine embryonic stem cells” by Reid et al. The authors exhibited that miRNAs play an important role of maintaining the intracellular Ca2+ homeostasis through the indirect regulation of Itpr2 expression level in ESCs. miRNAs might affect the Rbl2, Lats2, and Cdk1a whose miRNA is important in the differentiation and cell cycle regulation of ESCs.
This reviewer found this study is interesting, however, this reviewer has following major concerns.
1. Do these effects of loss of Dicer really represent loss of miRNAs? The authors showed only indirect evidence that miRNAs regulate Ca2+ homeostasis.
2. In Fig.3, the authors investigated apoptosis using a reduced concentration of thapsigargin at 200 nM. However, the authors did not examine that the effect of 200 nM thapsigargin on Ca2+ response. This should be confirmed.
3. In Fig.2A and 5A, the Ca2+ response to thapsigargin treatment in Dicer-/- ESCs appears to be faster than in Dicerfl/fl ESCs. The authors should mention about this.
4. The authors claim that thapsigargin-induced cytoplasmic Ca2+ increase from the ER. But mitochondria can also be a candidate for intracellular Ca2+ resource. Thapsigargin inhibits only Ca2+ uptake into the ER, does not promote to release Ca2+ from the ER. Why did not the authors consider the mitochondria or other resources?
Minor comments:
1. In the abstract, please add the explanation about Dicer.
2. In the introduction, it is necessary to describe in more detail the miRNAs that the authors focused on.
3. In L67-L69, please insert the reference.
4. In Fig.1A-C, did the authors perform the statistical analysis?
5. In the result section 3.2., the authors should more clearly mention that thapsigargin increased cytoplasmic Ca2+. This makes it easier for the reader to understand.
6. In Fig.5, the authors should mention how the extracellular Ca2+ was removed.
Minor editing of English language required.
Reviewer 2 Report
Overall, the paper contributes to our understanding of the regulatory role of miRNAs in Ca2+ homeostasis in murine embryonic stem cells. The findings shed light on the specific effects of miRNA deletion on Ca2+ response, apoptosis, and stress response pathways. The identification of Itpr2 as a target of miRNA regulation provides valuable insights into the mechanisms underlying Ca2+ signaling regulation in pluripotent stem cells.
Comments:
Figures: Please include the N numbers of repeated experiments in all figure legends.
Figure 1A: The legends were mislabeled. It should be "Left and right" instead of "above and below."
Figure 1E: Please explain why miR-19 is highlighted in yellow.
Section 3.4: This part of the results is not quite related to the work presented before or after. The authors need to provide further explanation regarding the source of the enhanced Ca2+ level upon adding ATP, whether it is from the extracellular space or the ER, and how this relates to Dicer depletion.
Round 2
Reviewer 1 Report
Please add a supplemental figure that the effect of 200 nM thapsigargin on Ca2+ response.
Author Response
We have added the figure as Figure S1, and we have modified the text in the results section with a sentence explaining the addition of the figure.